# Comparison Against Task Driven Artificial Neural Networks Reveals Functional Organization of Mouse Visual Cortex

**Jianghong Shi**
Department of Applied Mathematics
University of Washington
Seattle, WA 98195
jhshi@uw.edu

**Eric Shea-Brown**
Department of Applied Mathematics
University of Washington
Seattle, WA 98195
etsb@uw.edu

**Michael A. Buice**
Allen Institute for Brain Science
Seattle, WA 98109
michaelbu@alleninstitute.org

## Abstract

Partially inspired by features of computation in visual cortex, deep neural networks compute hierarchical representations of their inputs. While these networks have been highly successful in machine learning, it remains unclear to what extent they can aid our understanding of cortical function. Several groups have developed metrics that provide a quantitative comparison between representations computed by networks and representations measured in cortex. At the same time, neuroscience is well into an unprecedented phase of large-scale data collection, as evidenced by projects such as the Allen Brain Observatory. Despite the magnitude of these efforts, in a given experiment only a fraction of units are recorded, limiting the information available about the cortical representation. Moreover, only a finite number of stimuli can be shown to an animal over the course of a realistic experiment. These limitations raise the question of how and whether metrics that compare representations of deep networks are meaningful on these datasets. Here, we empirically quantify the capabilities and limitations of these metrics due to limited image presentations and neuron samples. We find that the comparison procedure is robust to different choices of stimuli set and the level of subsampling that one might expect in a large-scale brain survey with thousands of neurons. Using these results, we compare the representations measured in the Allen Brain Observatory in response to natural image presentations to deep neural network. We show that the visual cortical areas are relatively high order representations (in that they map to deeper layers of convolutional neural networks). Furthermore, we see evidence of a broad, more parallel organization rather than a sequential hierarchy, with the primary area VISp (V1) being lower order relative to the other areas.

## 1   Introduction

Deep neural networks, originally inspired in part by observations of function in visual cortex, have been highly successful in machine learning [14, 6, 21], but it is less clear to what extent they can provide insight into cortical function. Using coarse-grained neural activity from fMRI and MEG, it has been shown that comparing against task-driven DNNs provides insights for functional organization of primates' brain areas [7, 3]. At the single-neuron level, it has been shown that deep neural networks

with convolutional structure and a hierarchical architecture outperform simpler models in predicting single-neuron responses in primates' visual pathway [2, 24, 12, 23].

To understand the overall structure and function of cortex, we require models that describe both the population representation as well as single cell properties. Artificial neural network models such as convolutional networks discard complexity in individual units (compared to real biological neurons) but provide a useful structure to model large-scale organization of cortex, e.g. by describing the progressive development of specific feature response through successive layers of computation. Conversely, given an artificial network, we can use its patterns of response as a "yardstick" to assess the nature and complexity of representations in real neural networks. Naturally, such an assessment requires a metric for comparing representations and a suitable model for comparison. Here we choose such models from the family of convolutional networks. We aim to assess the complexity and hierarchical structure of a real cortical system relative to a computational hierarchy originally inspired by biological response.

Additionally we must choose a metric. While there exist metrics in the literature to compare representations between models or networks, even the largest scale neuroscience experiments only record from a fraction of the population of neurons and limited imaging or recording time implies that one can only cover a very small portion of stimulus space, raising the question of whether metrics that compare representations of deep networks to those of cortical neurons are meaningful. For example, the Allen Brain Observatory, despite being a massive dataset, includes only a small fraction of the neurons in the mouse visual cortex. Similarly, despite over three hours of imaging per experiment, only 118 unique natural images are shown due to the inclusion of a diverse array of stimulus types.

In this work, we empirically investigate the limitations imposed on representational comparison metrics due to limited presentations of stimuli and sampling of the space of units or neurons. Specifically, given a metric $M$ that computes a similarity score between two representations, we choose a fiducial task-trained network $X$ (such as VGG16 [21]) and ask about the robustness of mapping representations to depth in the network $X$ as a measure of feature complexity, we call this the $X$-pseudo-depth for metric $M$, $d_X^M$, of the representation. We use two metrics available in the literature, the similarity-of-similarity matrix (SSM) [5] and singular value canonical correlation analysis (SVCCA) [20, 17].

For both metrics, we compute the effect on VGG16-pseudo-depth and similarity score of the size of the image set and the number of units sub-sampled (as would happen, e.g. when a measurement precludes access to the entire population). We find that although the similarity score degrades with subsampling neurons, it can be well approximated with number of sampled neurons on the order of thousands. The pseudo-depth is also reasonably preserved with number of sampled neurons on the order of thousands.

Using these observations, we find that the data from the Allen Brain Observatory meets criteria that allow us to use the model VGG16 as a comparison model to assess functional organization and feature complexity via the similarity score and VGG16-pseudo-depth. We find that all regions of mouse visual cortex have the pseudo-depth close to the midpoint of the network, indicating that the representations as a whole are higher-order than the "simple" type of cell responses that typically used to describe early visual layers. The primary area VISp (also called V1) is of consistently lower VGG16-pseudo-depth than other layers, while the higher visual areas have no clear ordering, suggesting the fact that mouse visual cortex is organized in a broader, more parallel structure, a finding consistent with anatomical results [25]. VISam and VISrl have such low similarity scores that this may suggest an alternative function, i.e. a network trained on another task may yield more similar features.

## 2    Methodology

**Problem Formalization and definitions**    Define a "representation matrix" of a system $X$, $R_X \in \mathbb{R}^{n \times m}$, to be the set of responses of $m$ units or neurons to $n$ images. Choosing a set of images, we choose a model network and a similarity metric $M \in \{SSM, SVCCA\}$ and compute the **VGG16-pseudo-depth** as $d_{VGG16}^M = \mathrm{argmax}_{i \in \text{layers of VGG16}} M(R_X, R_{VGG16_i})$. We use $d^*$ as short hand notation for $d_{VGG16}^M$, and compute the corresponding **similarity score**, as $s^* = M(R_X, R_{VGG16_{d^*}})$. Our goal is to investigate the stability of $d^*$ and similarity score $s^*$ under subsampling of neuron number $n$ and both the number of images $m$ and which images are shown, and to use these quantities

to study representations across different mouse cortical areas. We also provide additional results about other model variants in the appendix.

**The Allen Brain Observatory data set**   The Allen Brain Observatory data set [4] is a large-scale standardized *in vivo* survey of physiological activity in the mouse visual cortex, featuring representations of visually evoked calcium responses from GCaMP6f-expressing neurons. It includes cortical activity from nearly 60,000 neurons collected from 6 visual areas, 4 layers, and 12 transgenic mouse Cre lines from 243 adult mice, in response to a range of of visual stimuli.

In this work, we use the population neural responses to natural image stimuli, which contains 118 natural images selected from three different databases (Berkeley Segmentation Dataset [16], van Hateren Natural Image Dataset [22] and McGill Calibrated Colour Image Database [18]). The images were presented for 250 ms each, with no inter-image delay or intervening "gray" image. The neural responses we use are events detected from $\Delta F/F$ using an L0 regularized deconvolution algorithm, which deconvolves pointwise events assuming a linear calcium response for each event and penalizes the total number of events included in the trace [10, 11]. Full information about the experiment is given in [4].

**Representation matrices for mouse visual cortex areas**   To construct the representation matrix for a certain mouse visual cortex area, we take the trial-averaged mean responses of the neurons in the first 500ms upon the image is shown. We group activities of neurons in different experiments for the same brain area and construct the representation matrix. Note that for the Allen observatory dataset, the number of images (118) is much less than the number of observed neurons.

**Representation matrices for DNN layers**   Unless explicitly stated, the representation matrices for DNN layers are obtained from feeding the same set of 118 images (resized to 64 by 64, see section 4.3 below) to the DNN and collecting all the activations from a certain layer.

**Two similarity metrics for comparing representation matrices**   We investigate two metrics suitable for comparing representation matrices with $n << m$, i.e., many fewer images than neurons. One is similarity of similarity matrices (SSM) [5]. Another is an extension of the recently developed singular value canonical correlation analysis (SVCCA) [20, 17] to the $n << m$ regime.

For the SSM metric, we calculate the Pearson correlation coefficient between every pair of rows in one representation matrix to get a size $n$ by $n$ "similarity matrix" where each entry describes the similarity of the response to two images. Importantly, this collapses the data along the neuron dimensions, so that representations with different numbers of neurons can be compared. To compare the two similarity matrices, we flatten the matrices to vectors and compute the Spearman rank correlation of their elements. Like the Pearson correlation coefficient, the rank correlation lies in the range $[-1, 1]$ indicating how similar (close to 1) or dissimilar (close to -1) the two representations are.

Following the established approaches [20], we first run singular value decomposition (SVD) to reduce the neuron dimension to a fixed number $r$ which is smaller than the dimension of both representations. We fix $r$ to be the most important (largest variance) 40 dimensions for each representation. We then perform a canonical correlation analysis (CCA) on the reduced representation matrices. CCA compares two representation matrices by sequentially finding orthogonal directions along which the two representations are most correlated. We can then read out the strength of similarity by looking at the values of the corresponding correlation coefficients. We take the mean of the $r$ correlation coefficients resulting from CCA as the SVCCA similarity value.

Note that SVCCA is invariant to invertible linear transformations of the representations. SSM is invariant to transformations of representations that induce monotonic transformations of the elements of similarity matrices. An excellent review of similarity metrics and their properties can be found in [13].

## 3   Robustness of estimates of similarity score and pseudo-depth to subsampling of images and neural units

In this section, we study the robustness of VGG16-pseudo-depth and similarity score estimates in the face of limited stimuli and limited access to neurons in the representation of interest. Recall that we have full access to all neurons in the pretrained VGG network [21] that we are using as a "yardstick.' We begin with the simplest possible setting: using this yardstick to measure VGG16-pseudo-depth

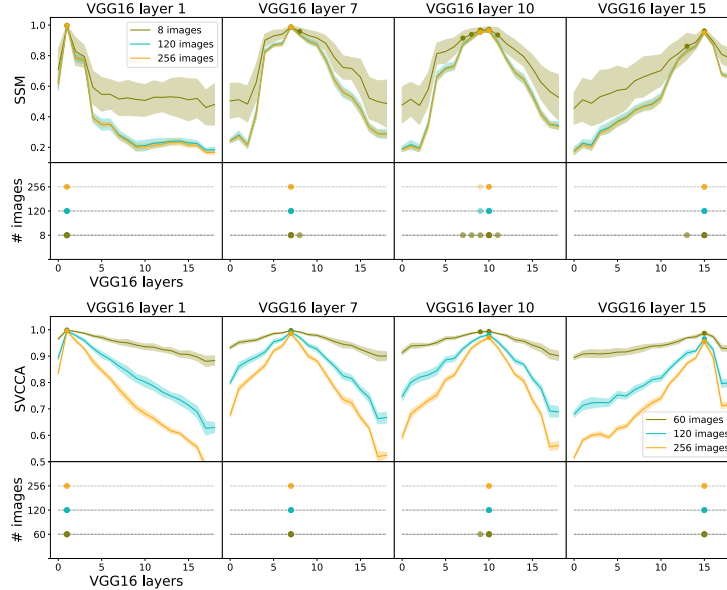

Figure 1: Testing the self-consistency of $d^*$ by varying the number of images included in the dataset. Shown are SSM (top) and SVCCA (bottom) $d^*$ computed for several layers of VGG16 (1, 7, 10, 15 from left to right) using different numbers of stimuli from tiny ImageNet. The shaded areas denote the standard deviation computed from different randomly chosen sets of images. The shaded circles denote the layers indistinguishable from $d^*$ (highlighted).

and similarity score of another copy of VGG16, but for which we observe only a random subsample of units (neurons).

We will show that (1) the similarity scores are robust to including only the 118 images in the Allen brain observatory data set, as well as the specific images within this set, and (2) the similarity scores decrease with neuron subsampling, whereas the pseudo-depth stays constant given enough neurons.

## 3.1 VGG-pseudo-depth and similarity scores can be estimated stably with limited image sets

The Allen Brain Observatory dataset includes neural responses to 118 natural image stimuli. We first study how the number of stimuli influences estimates of VGG16-pseudo-depth and similarity score, and how much variation arises when we present different sets of images.

For this, we randomly select different numbers of images from tiny ImageNet and calculate the similarity values between VGG16 model layers. The results for four representative layers are shown in Figure 1. We see that the VGG16-pseudo-depth identifies the corresponding layer that is chosen for comparison, and the similarity score is always one for the corresponding layer given different number of randomly chosen images. In addition, the variance introduced by the random choices of images is small for 120 images. Thus the metrics are robust to different choices of stimuli set, presumably including the image set used in the Allen Brain Observatory.

Note that the sharpness of the peak of the similarity curve represents how much the metric can differentiate one layer from another. We see that for SSM layers do not further differentiate when more than 120 images are shown (approximately the number presented in the biological data set), while SVCCA values can still differentiate the layers better, with more peaked similarity curves, if we add more images to the data set.

## 3.2 VGG-pseudo-depth and similarity scores can be estimated stably with sufficient subsampling of neuronal populations

In biological experimental settings, we only observe a small portion of neurons from a brain area. Here, we investigate how this affects our ability to reliably use the VGG network to estimate pseudo-depth and similarity scores. Recalling that the network that we use as a yardstick can be completely observed, we take a sub-subsampled population from a certain layer in VGG16, and compare it to the whole population of VGG16 layers. The results for four representative layers are given in Figure

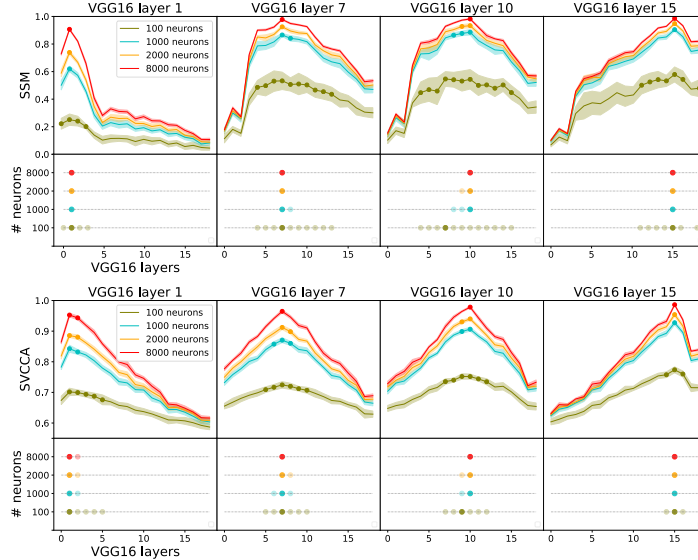

Figure 2: Testing the self-consistency of $d^*$ by varying the number of units subsampled. Shown for SSM (top) and SVCCA (bottom) is $d^*$ computed for several layers of VGG16 (1, 7, 10, 15 from left to right). The shaded areas denote the standard deviation computed from different random draws of sub-samples.

2. This shows that the similarity scores are severely reduced by subsampling. As we increase the number of neurons, the similarity curves also rise, reaching values with 2000 neurons that are close to those with no subsampling. Thus, at least for comparing the VGG model with a partially observed version of itself, a rule of thumb is that if including at least 2000 neurons in the sampled population, then the similarity score is a good approximation to those that would be found from observing the whole population.

The relative order of similarity values across layers is consistent for a wider range of the number of sampled neurons. Even with less than 2000 neurons sampled, say 1000, we can already find which layers are more similar to the population of interest. Thus, the corresponding rule of thumb for VGG16-pseudo-depth is that around 1000 neurons must be sampled for it to be consistently estimated.

### 3.3 Robustness of similarity score and pseudo-depth extend to a different network

To see whether the approaches above remain robust when comparing representations from a different network against representations generated by VGG16, we choose neurons from 4 layers of VGG19 and compare them with entire layers of VGG16. The results are given in Figure 3. We see that the curves with 2000 neurons are very close to the ones with 8000 neurons, suggesting that this remains an adequate level of sampling when comparing between these two networks. Moreover, our metrics show that early layers in VGG19 are more similar to early layers in VGG16, and later layers in VGG19 are more similar to later layers in VGG16, as we would expect intuitively, reflecting the functional hierarchy of the four VGG19 layers based on VGG16-pseudo-depth estimated from 2000 neurons.

## 4 VGG16-pseudo-depth and similarity scores for mouse cortex and interpretations for the visual hierarchy

In this section, we compare mouse visual cortex representations against VGG16 and discuss the resulting insights for the mouse visual hierarchy. In the Allen brain observatory data set, each neuron belongs to a specific visual area (VISp, VISl, VISal, VISpm, VISal, VISam), cortical layer (layer23, layer4, layer5, layer6) and has a specific cell type (Cre-line). By grouping neurons in different areas, cortical layer or cell types, we can study the functional properties of the specific neuron groups. In the following, we separately compare VGG16 with entire cortical areas (Figure 4), distinct cortical layers in the same area (Figure 5), and distinct cell types in the same area (Figure 6).

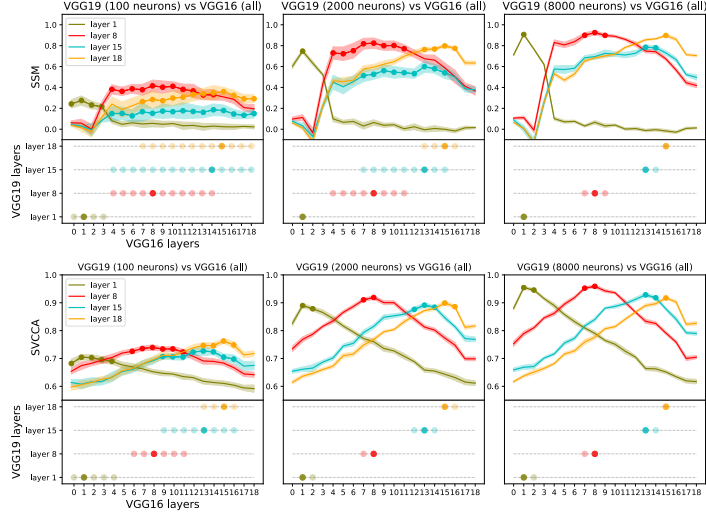

Figure 3: $d^*$ computed on the layers of VGG19. $d^*$ is relatively consistent across with large numbers of sub-sampled units. Shown for SSM (top) and SVCCA (bottom) is $d^*$ for the layers of VGG19 with different numbers of sub-sampled units (left to right: 100, 2000, 8000).

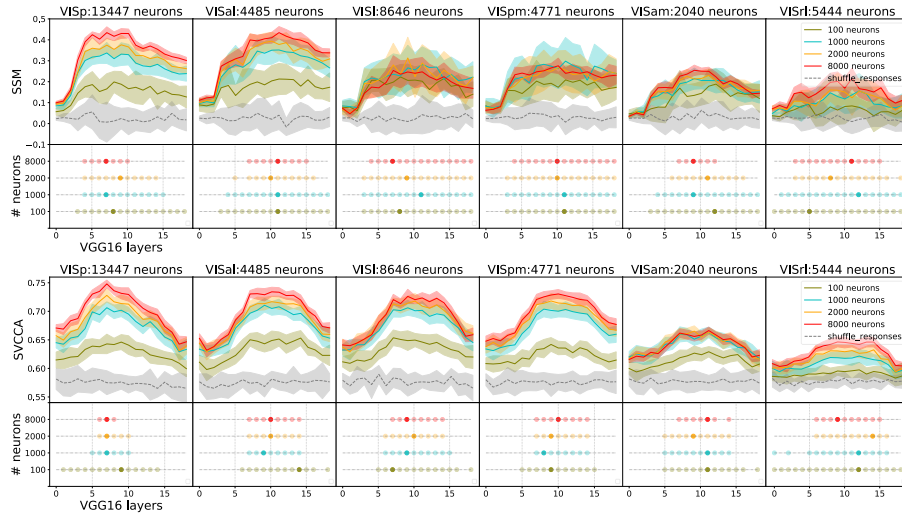

Figure 4: $d^*$ computed for representations from the Allen Brain Observatory, shows a relatively broad, parallel structure, rather than a strict hierarchy, although VISp is of lower $d^*$ than the other areas. Shown for SSM (top) and SVCCA (bottom) is $d^*$ for the Allen Brain Observatory. The dashed gray curve is comparing the whole population to VGG16 with responses shuffled. The shaded areas denotethe standard deviation computed from different random draws of sub-samples. The shaded circles denote the layers indistinguishable from $d^*$ (highlighted).

## 4.1 Whole brain area comparisons show functional properties for mouse visual cortex areas

To study visual representations within and across whole brain areas, we group all the neurons in the same visual area and compare all six areas in our data set to VGG16. Note that different areas have different total numbers of recorded neurons available. In order to make fair comparisons across areas, each time we compute a similarity curve we sample the same number of neurons with replacement from each area. As always, we compare representations in the sub-sampled brain area to representations in all neurons in the VGG16 layers that we are using as our yardstick. The results are shown in Figure 4. To give a baseline for these comparisons, we shuffled the rows of the representation matrices and calculate the similarity curves for it (dashed gray curves).

Similarity curves computed using both SSM and SVCCA metrics show that:

1. The pseudo-depth for the mouse brain areas corresponds to the middle layers of VGG16. This shows that mouse visual cortical representations are *higher-order*, involving multiple stages of processing.

2. The pseudo-depth of VISp is lower than that of other brain areas, a fact that is partially but not completely attributable due to its receptive field size (see section 4.3 below). Meanwhile, the higher visual areas have no clear ordering. This suggests that, following initial stages of processing after VISp, mouse visual cortex is organized in a broadly parallel structure, as apposed to a hierarchical one.

3. VISam and VIrl have the lowest similarity scores among all brain areas, according to both metrics. Based on our studies in Section 3 (Figure 2) that suggest sub-samples of 2000 neurons are sufficient to approximate similarity scores, this indicates that VISam and VIrl are less similar overall to VGG16 than the other areas. A natural hypothesis is that VISam and VIrl perform a different type of processing – one that demands visual features that are more distinct from those required to classify the large set of categories used to train VGG16.

In addition to these principal observations that are common to both SSM and SVCCA metrics, we note that these metrics do show some different properties when applied to brain areas VISl and VISpm. Specifically, SSM produces a relatively larger variance in similarity curves across subsamples of VISl and VISpm neurons, and as a consequence a broader range of possible pseudo-depths for these areas. We leave investigating the cause and possible interpretations of such differences to future work. We also used different input image resolutions to do the comparison (Figure 7 in appendix), it shows our main conclusions remain valid, but increasing image resolution cause the pseudo-depth to shift to the right, which suggests that the pseudo-depth could be associated with receptive field size. To numerically quantify the effects of trial-to-trial variability, we repeated the calculation of the SSM value as in Figure 4 by bootstrapping across trials (Figure 8 in the appendix). The results show that our main conclusions are robust to trial-to-trial variability.

## 4.2 Cortical layer and cell-type subpopulations show similar trends but can have higher similarity scores than brain areas taken as a whole

How do the trends for pseudo-depth and similarity scores that we have identified above depend on the fact that we have grouped together neurons across type and cortical depth (*cortical* layer) into 'whole" areas? To answer this question, we separate neurons from the same brain area according to their cortical layer, Figure 5, and genetically encoded cell line (a coarse measure of cell type), Figure 6. In producing the resulting similarity scores we sample 2000 neurons with replacement from each subpopulation of mouse neurons. Note that these subpopulations have less than 2000 neurons in general, so that resampling is significant; in Figure 6, we only show the results for cell types with more than 900 neurons.

We find that SVCCA reveals the same basic trends in similarity curves when brain areas are divided into subpopulations as for the whole area comparisons in Section 4.1. The SSM metric produces curves that are suggestive of some possible differences. For example, for the whole area comparisons, we see that the SSM curves values for VISl and VISpm have lower mean and larger variance compared to those for VISp and VISal. However, when their subpopulations are considered separately, there are some cortical layers (layer23 of VISl and layer5 of VISpm) and cell types (Slc17a7 of VISpm) that have higher SSM similarity scores than their areas as a whole. This suggests that these cortical layers and cell types may, taken as components of a larger system, represent visual features that are in fact more similar to those extracted by VGG16.

## 4.3 Impact of image resolution on VGG pseudo-depth

A natural question about our conclusions about pseudo-depth above is whether they are an automatic consequence of the image resolution (sometimes referred to as the receptive field size) that occurs at different stages through both the VGG network and the mouse brain – in other words, whether they simply follow from matching the resolution in a given VGG layer with that in a given mouse brain area, rather than matching their complexity.

To address this, we first note that we have chosen our input images to be downsampled to a very limited size (64 by 64) that roughly corresponds to the limited visual acuity of the mouse [19]. Thus we do not believe that our overall finding that the VGG-pseudo-depth of mouse visual brain areas

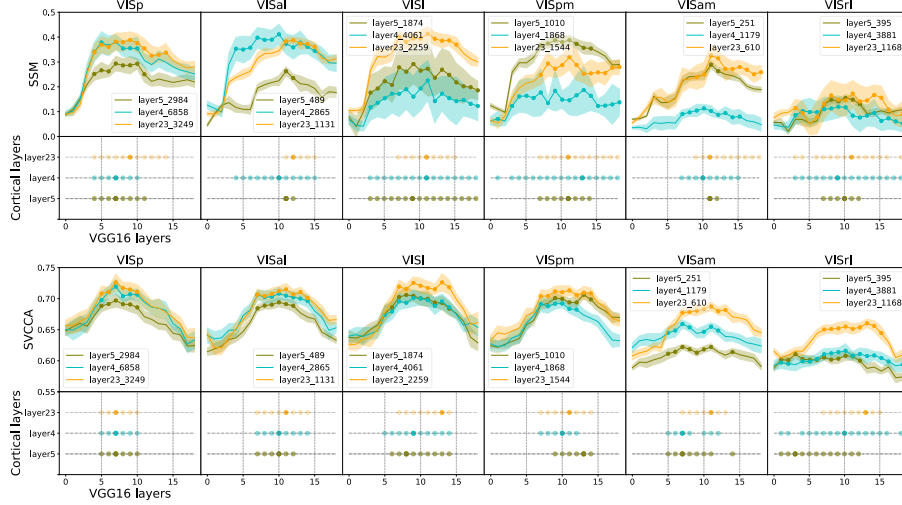

Figure 5: Separate cortical layer comparisons. SSM (top) and SVCCA (bottom) result for comparing different cortical layers in the same area to VGG16.

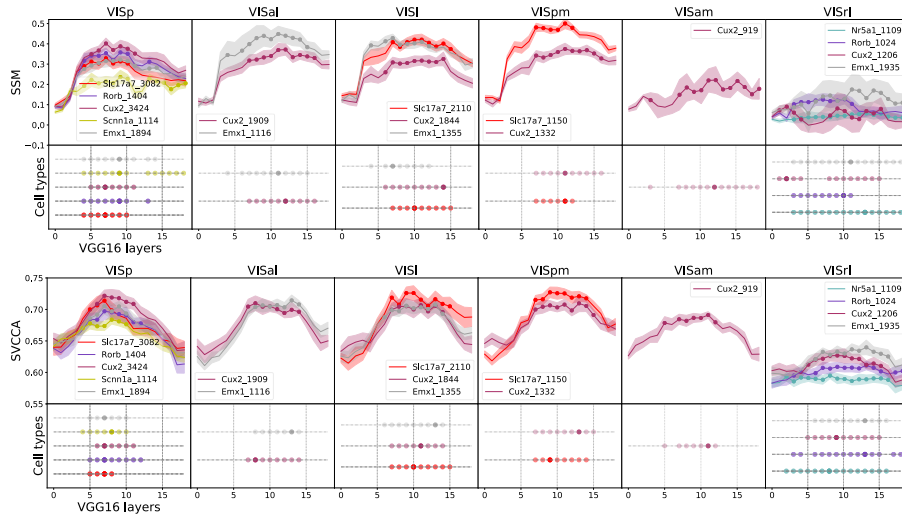

Figure 6: Separate cell type comparisons. SSM (top) and SVCCA (bottom) result for comparing different cell types in the same area to VGG16. Only cell types with more than 900 neurons are shown.

corresponds to the middle layers of VGG is an automatic consequence of needing to look sufficiently deep into the VGG network for receptive field sizes that are as large as those in the mouse visual system. In the appendix, we further test this by recomputing similarity curves for the VGG network responding to images with both substantially lower (resized input images to 32 by 32) and higher (128 by 128) resolution. We find that there is little effect of this input resolution for SSM pseudo-depth. Moreover, while SVCCA pseudo-depth is impacted by input resolution, pseudo-depths remain in the middle layers of SVCCA even when the input resolution is doubled or halved. Based on this we conclude that our result that the pseudo-depth of mouse visual cortex corresponds to the middle layers of VGG16 is robust to reasonable assumptions about the visual resolution. However, conclusions about the *relative* depth of visual areas could still be impacted by the resolution issue. For example, area VISp is known [4] to have smaller receptive fields than other mouse visual cortex areas. Thus, the fact that SVCCA (but not SSM) pseudo-depths are earlier for VISp than other areas could be due to the resolution effects, rather than the level of complexity of its representations. We note a final possible limitation in interpreting our results. The VGG16 network that we use as a yardstick was pretrained on high resolution visual inputs. It is an interesting and open question whether our findings would be the same for a network retrained with the lower resolution inputs which we use and describe above.

# 5    Conclusion

Deep artificial neural networks can now produce task behavior that rivals the performance of biological brains in many settings. This opens the door to a fascinating question: what is similar, and what is different, in the way in which artificial and biological networks solve the underlying tasks [23, 9]. A natural place to start is in comparing the stimulus representations that each produces.

Our first goal was to assess the robustness of this comparison to an unavoidable challenges: the set of stimuli, and number of neurons, that can be probed in biological experiments is necessarily limited. Our empirical results show that pseudo-depth and similarity scores are indeed robust to choices of stimuli on the order of hundreds and subsampling of neurons on the order of thousands.

Our second goal was to use this comparison to investigate visual representations in the mouse visual cortex, a system of explosively increasing interest in the neuroscience community and for which curated, massive public data sets on visual representations are now available [4]. Functionally, very little is known about the visual areas in mice, compared with the primate visual cortex. This said, anatomical studies are developing the inter-area wiring diagram ([8]), and functional studies have provided evidence of some specialization across areas in terms of spatial and temporal frequency processing (e.g. [1, 15]). Our results with data from the Allen Brain Observatory data set show that, according to VGG pseudo-depth and similarity scores, mouse visual cortical areas are relatively high order representations in a broad, more parallel organization rather than a sequential hierarchy, with the primary area VISp being lower order relative to the other areas. This is consistent with the relatively flat hierarchy observed in [8]. This approach and finding invites future insights from other artificial network systems, e.g. recurrent networks, and helps open doors for analyzing emerging large-scale datasets across species and tasks.

# 6    Acknowledgements

We thank Tianqi Chen, Saskia de Vries, Michael Oliver for helpful discussions, and Rich Pang, Gabrielle Gutierrez for comments on the draft. We thank the Allen Institute for Brain Science founder, Paul G. Allen, for his vision, encouragement, and support. We acknowledge the NIH Graduate training grant in neural computation and engineering (R90DA033461).

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
