[Supplementary Material]

# Appendix

## A   An image size control shows that our main conclusions remain valid for input images with different scales and resolution

Figure 7: SSM (top) and SVCCA (bottom) result for comparing mouse visual cortex areas with 2000 sample neurons to VGG16 with different resizing (32x32, 64x64, 128x128) of input images. The shaded areas are the standard deviation computed from different random draws of sub-samples. The shaded circles denote the layers indistinguishable from the layer with highest similarity (highlighted).

## B   VGG16-pseudo-depth of mouse brain areas with bootstrapping across trials

Figure 8: Same comparison as in Figure 4, with shaded area showing standard deviation from simultaneously performing bootstrap resampling for trials' mean responses and random draws of different sub-samples of neurons.

## C   SSM results for AlexNet, ResNet18, VGG16 variants

In the following, we repeat all the experiments in the main paper for AlexNet, ResNet18, and two Pytorch VGG16 models (VGG16a and VGG16b) trained from different initializations. The VGG16 model in the main paper is a pre-trained Tensorflow model, which has a different preprocessing scheme from the Pytorch model. The type of each numbered layer, for each model, is given in Table 1. These results show that our main conclusions are preserved by models with different architectures or initializations.

Figure 9: Same experiments as Figure 1 for AlexNet, ResNet18, VGG16a, VGG16b (top to bottom). These results show that, as for the VGG16 model in the main paper, 120 images is generally an adequate number to identify layers via SSM for the other models as well.

Figure 10: Same experiments as Figure 2 for AlexNet, ResNet18, VGG16a, VGG16b (top to bottom). These results show that, as for the VGG16 model in the main paper, subsampling neurons on the order of thousands will give a good approximation of SSM similarity values.

Figure 11: Same experiments as Figure 3 for AlexNet, ResNet18, VGG16a, VGG16b (top to bottom). These results show that, as for the VGG16 model in the main paper, other models can be used as yardsticks to differentiate VGG19 layers via SSM as well.

Figure 12: Same experiments as Figure 4 for AlexNet, ResNet18, VGG16a, VGG16b (top to bottom). These results show that our main conclusions about mouse visual cortex are qualitatively preserved by different models.

Figure 13: Same experiments as Figure 5 for AlexNet, ResNet18, VGG16a, VGG16b (top to bottom). These results show that our main conclusions about mouse visual cortex are qualitatively preserved by different models.

Figure 14: Same experiments as Figure 6 for AlexNet, ResNet18, VGG16a, VGG16b (top to bottom). These results show that our main conclusions about mouse visual cortex are qualitatively preserved by different models.

Table 1: Specification of layers for each network model.

|  | VGG16 | VGG19 | ResNet18 | AlexNet |
|---|---|---|---|---|
| 0 | Input | Input | Input | Input |
| 1 | Conv | Conv | bn1 | Conv |
| 2 | Conv | Conv | Maxpool | Maxpool |
| 3 | Maxpool | Maxpool | layer1.0.bn1 | Conv |
| 4 | Conv | Conv | layer1.0.bn2 | Maxpool |
| 5 | Conv | Conv | layer1.1.bn1 | Conv |
| 6 | Maxpool | Maxpool | layer1.1.bn2 | Conv |
| 7 | Conv | Conv | layer2.0.bn1 | Conv |
| 8 | Conv | Conv | layer2.0.bn2 | Maxpool |
| 9 | Conv | Conv | layer2.1.bn1 | |
| 10 | Maxpool | Conv | layer2.1.bn2 | |
| 11 | Conv | Maxpool | layer3.0.bn1 | |
| 12 | Conv | Conv | layer3.0.bn2 | |
| 13 | Conv | Conv | layer3.1.bn1 | |
| 14 | Maxpool | Conv | layer3.1.bn2 | |
| 15 | Conv | Conv | layer4.0.bn1 | |
| 16 | Conv | Maxpool | layer4.0.bn2 | |
| 17 | Conv | Conv | layer4.1.bn1 | |
| 18 | Maxpool | Conv | layer4.1.bn2 | |
| 19 | | Conv | Avgpool | |
| 20 | | Conv | | |
| 21 | | Maxpool | | |