[Reviews · NeurIPS 2019]

Reviewer 1



Strengths: I found the authors’ formulation of network pseudo-depth to be a very interesting and potentially useful metric for comparing artificial neural network models to neural data. I think their finding (displayed in Figure 2) that the number of sampled neurons had to be around at least 1000-2000 for the VGG-16 pseudo-depth to be consistently estimated, and that this finding holds when comparing representations against another network (VGG-19), demonstrates a potentially useful rule-of-thumb for adequate population sizes in neural data. Furthermore, their finding that mouse visual cortex is more parallel after a few stages of hierarchical processing starting at around area VISp, could be useful for building better task-driven models of mouse visual cortex, and indicates an important distinction with the traditional, hierarchical primate ventral visual pathway. Weaknesses: I would have liked to see more analyses of the robustness of the pseudo-depth metric with different networks, especially those not in the VGG family. I am aware that the Allen Institute has compared VGG-16/19 to their mouse data, and therefore, this is likely why the authors chose this model to begin with. However, I don’t think the Allen Institute’s choice of model is particularly well-motivated, and given both the anatomy and the results of this paper, likely not a very good model of mouse visual cortex anyway (not to mention not a very good ImageNet model either). I would have liked to see comparisons with shallow ResNet models or even AlexNet. Furthermore, while their current analysis of the pseudo-depth metric they introduce seems to indicate needing at least thousands of neurons in order to make robust quantitative comparisons between features, it is not clear to me what further neuroscience knowledge we have gained beyond what we expected from the anatomy of mouse visual cortex (namely, that these pathways are more parallel than hierarchical). Given that they are finding that the pseudo-depth of mouse visual cortex corresponds roughly to the middle layers of a VGG-16, it would have seemed more relevant to compare an AlexNet or similar shallow convolutional neural network architecture to their neural data and for their analysis of the robustness of pseudo-depth. Without offering a potentially better solution for artificial neural network models of mouse visual cortex beyond what has already been known (e.g. using a VGG-16/19 and having their findings agree with rough anatomical considerations of mouse visual cortex), their only novel finding is the use of their pseudo-depth metric and their finding that thousands of neurons are necessary to make robust comparisons (with the caveat that they only tried this on two neural networks). I think this result on its own is interesting.

Reviewer 2



Updated after authors' response ------------------------------------------- Thanks to the authors for their well-written response. In my original review, I failed to fully appreciate the novelty and significance of the result that mouse cortical areas are not organized hierarchically. In addition, I appreciate the added background on mouse cortical areas (as well as the pointer about VISp, I apologize for missing that detail in my original reading!). In addition, I appreciate the response to my comments about audience. Comparisons between artificial networks and biological data are of growing importance, as such NeurIPS is definitely an appropriate venue for this work. After reading their response, and the other reviews, I have decided to increase my score to a 7. Original review -------------------- Originality: The work is largely original. I would point the authors to a (recent) paper that addresses a related question (comparing different methods for measuring representational similarity): Similarity of neural network representations revisited. Kornblith et. al. ICML 2019. Kornblith et. al. propose a similar test for assessing the quality of a similarity metric (comparing all of the layers of a network to a given layer of a network with the same architecture). One key difference is that the Kornblith et. al. paper compared networks with the same architecture trained using different random initializations, whereas in this paper the authors use the exact same network (pre-trained). Quality: The work is fairly thorough. I appreciated the comparisons to two popular metrics (SSM and SV-CCA), as well as providing a sense of the spread in the results due to random sampling by adding shaded error bars (side note: are these standard error or standard deviation, or something else?). I would like to have seen comparisons with multiple CNN architectures (not just VGG), as well as comparisons within the same architecture but trained with different random seeds. The authors note this potential caveat of their findings in the conclusion, namely that their results are only for comparisons to VGG itself. I think this caveat is significant enough that it should be addressed in this paper. Regarding the second half, I wish the authors could have also compared with mouse V1, rather than just the higher cortical areas. Mouse V1 would provide a nice baseline, as we would expect it to be most similar to early layers of convolutional networks. (note: I am not sure if this type of mouse V1 data is available in the Allen dataset, so this suggestion may be difficult to try). Clarity: The paper is largely well written and clear. I would have appreciated more discussion and interpretation of the scientific findings. In particular, there was little to no background on what the different mouse cortical areas are, and what is currently known about them. Due to this lack of background, the scientific motivation driving the second half of the paper is weak, and it is hard to know what to make of those results. Finally, I would have appreciated some discussion on similarities/differences of the two metrics (SSM vs SVCCA). Do the authors prefer one over the other, given their findings? Significance: The first part of the paper addresses an important question, relevant for the NeurIPS audience (when should we trust comparisons between network representations?). However, some of the methodological limitations (only studying VGG, not re-training with different random initializations) reduce the overall significance of the work. The second part of the paper tries to address a scientific question (how do representations in mouse visual cortex compare to VGG16?) but this question is not well motivated, and there is a lack of background material presented to put these findings in context. For example, it is unclear to me if the finding that 'mouse visual cortical areas are relatively high order representations in a broad, more parallel organization' is all that new or surprising. I think these limitations reduce the significance of the second half of the paper.

Reviewer 3



Review update: Thanks for your responses. I still think it's a good paper. Nice work! Just one piece of feedback on the rebuttal: You promise to run more analyses and include them in the paper. I wish you would have indicated some of the results of these analyses in the rebuttal. That would have made it stronger. Summary The paper compares the effect of various variables such as image number, neuron number, and image resolution on comparisons of representational similarity between different mouse brain regions and deep network layers. After concluding that the data from Allen brain observatory meet the criteria for a robust comparison, they perform the analysis on that data and conclude that there is little evidence for a hierarchy in mouse visual cortex. Strengths and weaknesses The paper adds to a series of comparisons of representational similarity and complexity using neuronal recodings and task trained deep networks. The paper is well written, an extensive set of analyses are performed. I am very sympathetic from this line of research and my experience with similar data aligns with their conclusions. Since we still know very little about visual representations in mouse visual cortices, the work is interesting and relevant. My only concern with the paper is that all the robustness comparisons were performed on deep networks and not neuronal data. The problem with that is that neuronal data is much less reliable than deterministic activities from deep networks. In addition mouse neuronal data suffers from non-visual non-stationarities caused by running activity or different states of alertness. Finally, 2-photon data is usually deconvolved (is that the case here?). Deconvolution is unable to identify scaling in the data. While this does not affect the SSM score, I wonder how this affects the SVCCA. In conclusion, the analysis would be more convincing if the authors could investigate how the above factors affect the robustness of the analysis. Minor comments - "Thus the metrics are robust to different choices of stimuli set, presumably including the image set used in the Allen Brain Observatory." Why not use exactly those stimuli? - How many images where used to compare the neuronal subsampling curves? - Can you provide a few more number on the experimental conditions under which the data was recorded (i.e. presentation time; scan rate; calcium indicator; how many animals is the data from, etc.)

[Author Response · NeurIPS 2019]

We thank the reviewers for their constructive feedback and their appreciation of our experiments on this paper. The
suggestions about adding more model variations are helpful; we will include more model variations (Alexnet, ResNet,
and different initializations) in the revision. We have already performed comparisons of many architectures with the
mouse data and achieve qualitatively similar results. We address some of the specific comments below.

**Reviewer 1**

***novelty of findings*** We appreciate the reviewer's comments about the pseudo-depth metric and criteria for its robustness
as novel findings. Our aim is to add another to the list: that we have used these tools to evaluate the *complexity* of
visual processing in mouse cortex. Our analysis with VGG networks in the submission shows that even the primary
area VISp (also called V1) computes substantially higher-order features than are commonly assumed in biological
models of visual processing (such as HMAX); in our revision we will, as the reviewer rightly requests, evaluate this
with shallower networks as well.

**Reviewer 2**

***compare to mouse V1*** VISp is an alternative nomenclature for V1, mentioned (we admit, briefly) in the text (L70). We
will emphasize it more in the revision. This confusion coincidentally illustrates that "mouse visual cortical areas are
relatively high order representations (including VISp) in a broad, more parallel organization" is, in fact, a surprising
result. Our results show that even the earliest stage of the mouse visual cortex computes more higher-order features
than is commonly assumed. This is consistent with a growing body of literature.

***shaded error bar*** These are standard deviation. We will clarify this in the revision.

***scientific background*** Functionally, very little is known about the visual areas in mice; we have nowhere near the level
of detail we have about the primate cortex. There is some knowledge from anatomy (Harris, et al, 2019). There is
some evidence for functional specialization in terms of spatial and temporal frequency processing (Andermann, et al,
2002, Marshel, et al 2011). In the absence of such information, the VGG16 pseudo-length gives us a window into
the functional organization of the higher visual areas, one that is roughly consistent with the relatively flat hierarhcy
observed in Harris, et al. We will add this discussion in the revision.

***SSM vs SVCCA*** They are two different metrics that have different properties. SVCCA is invariant to affine transfor-
mation on the features. SSM is invariant to monotonic transformation on the similarity matrices. We would therefore
advocate using either depending upon the precise question. We will add this discussions in the revision.

***NeurIPS audience*** This work provides a robust method that reveals the functional organization of a biological visual
system (mouse) whose neural coding properties are currently – and relatively recently – the subject of very intense
study across computational neuroscience. This is important, because data are available for the mouse on a totally
unprecendented scale – which, as we show here, enables new questions to be asked. It is also important because this
system appears to show, even in its early areas, distinctly higher-order coding properties when compared with the
standard view of, say, the macaque ventral stream. Thus, our work provides new inroads at the interface between
engineered and biological computing networks that have long been a mainstay of NeurIPS.

***Kornblith et. al. ICML 2019.*** Thanks for pointing out this reference. Kornblith et. al. discussed the properties of the
similarity metrics on comparing artificial neural networks. Our paper focuses on robustly studying systems in which
one does not have access to all units, as at present must be the case for biological systems such as mouse visual cortex.
We will add a citation in the revision.

**Reviewer 3**

***2-photon data is usually deconvolved*** The data is deconvolved by the algorithm in (Jewell et. al. 2018). Full information
about how the data was processed is given in the Allen Institute paper de Vries, et al 2018. We will provide more
information about the data in the revision.

***deconvolution is unable to identify scaling*** SVCCA is invariant to affine transformation on the features, thus is also
not affected by the scaling in the data.

***trial-to-trial variability and non-stationarity*** This is a very important question. We will add results of bootstrapping
trials to quantify the effects of trial-to-trial variability and non-stationarity among trials in the revision.

***why not use exactly those stimuli*** The primary reason we use images other than those used for the Brain Observatory
are so that we can study the sampling trends beyond 118 images (the number shown in the experiment). For the
comparisons that don't require this, we do use the exact stimuli (such as when comparing to the mouse data), including
for the neural subsampling curves.

[Meta-Review · NeurIPS 2019]

Dear authors, congrats on the acceptance-- this paper was discussed extensively, the the reviewers provided multiple comments and feedback-- please do take the feedback in the reviews into account when preparing your final manuscript..